# Microvasculature Features Derived from Hybrid EPI MRI in Non-Enhancing Adult-Type Diffuse Glioma Subtypes

**DOI:** 10.3390/cancers15072135

**Published:** 2023-04-04

**Authors:** Fatemeh Arzanforoosh, Sebastian R. van der Voort, Fatih Incekara, Arnaud Vincent, Martin Van den Bent, Johan M. Kros, Marion Smits, Esther A. H. Warnert

**Affiliations:** 1Department of Radiology and Nuclear Medicine, Erasmus MC, 3015 GD Rotterdam, The Netherlands; 2Brain Tumor Center, Erasmus MC Cancer Institute, 3015 GD Rotterdam, The Netherlands; 3Department of Neurosurgery, Erasmus MC, 3015 GD Rotterdam, The Netherlands; 4Department of Neurology, Erasmus MC, 3015 GD Rotterdam, The Netherlands; 5Department of Pathology, Erasmus MC, 3000 CB Rotterdam, The Netherlands; 6Medical Delta, 2629 JH Delft, The Netherlands

**Keywords:** vessel size imaging, microvessel, perfusion imaging, cerebral blood volume, glioma, molecular typing

## Abstract

**Simple Summary:**

Reliable insight into tumor microvasculature is important for monitoring disease progression and treatment response. Vessel size imaging (VSI), an emerging MRI technique, has shown great potential in revealing accurate information about microvasculature in gliomas. However, the technique has rarely been used in the clinical setting, as investigated here. Of note, our work was only aimed at non-enhancing tumors. This sets our work apart from other studies investigating tumor microvasculature, typically including enhancing tumors, i.e., tumors in which angiogenesis is already evident macroscopically. Additionally, our findings are of potential clinical relevance for differentiating the most aggressive tumor subtype, glioblastoma, from two other glioma subtypes: oligodendroglioma and astrocytoma. This is particularly challenging when there is no contrast enhancement.

**Abstract:**

In this study, we used the vessel size imaging (VSI) MRI technique to characterize the microvasculature features of three subtypes of adult-type diffuse glioma lacking enhancement. Thirty-eight patients with confirmed non-enhancing glioma were categorized into three subtypes: Oligo (IDH-mut&1p/19q-codeleted), Astro (IDH-mut), and GBM (IDH-wt). The VSI technique provided quantitative maps of cerebral blood volume (CBV), microvasculature (µCBV), and vessel size for each patient. Additionally, tissue samples of 21 patients were histopathologically analyzed, and microvasculature features were quantified. Both MRI- and histology-derived features were compared across the three glioma subtypes with ANOVA or Kruskal–Wallis tests. Group averages of CBV, μCBV, and vessel size were significantly different between the three glioma subtypes (*p* < 0.01). Astro (IDH-mut) had a significantly lower CBV and µCBV compared to Oligo (IDH-mut&1p/19q-codeleted) (*p* = 0.004 and *p* = 0.001, respectively), and a higher average vessel size compared to GBM (IDH-wt) (*p* = 0.01). The histopathological analysis showed that GBM (IDH-wt) possessed vessels with more irregular shapes than the two other subtypes (*p* < 0.05). VSI provides a good insight into the microvasculature characteristics of the three adult-type glioma subtypes even when lacking enhancement. Further investigations into the specificity of VSI to differentiate glioma subtypes are thus warranted.

## 1. Introduction

Angiogenesis is an essential process for glioma development and growth [1]. In addition to increased vessel density, it also leads to morphologically abnormal vessels compared to healthy tissue [2]. As the tumor becomes more aggressive, it develops a highly defective vasculature with tortuous vessels of variable diameter and abnormal capillary bed topology, arteriovenous shunting, and microthromboses [3,4]. At this stage, angiogenesis becomes visible with MRI in the form of contrast enhancement. The histopathological features of glioma vasculature, i.e., piling up of endothelial cells, pericytes, and smooth muscle positive mural cells, are traditionally used for predicting tumor malignancy [5]. 

Three main subtypes of adult-type diffuse glioma are recognized in the 2021 WHO classification of tumors of the central nervous system [6]. These subtypes are: Oligodendroglioma with IDH mutation and 1p/19q-codeletion, Oligo (IDH-mut&1p/19q-codeleted); Astrocytoma with IDH mutation but without 1p/19q-codeletion, Astro (IDH-mut); and Glioblastoma without IDH mutation, i.e., IDH wildtype, GBM (IDH-wt). Among these subtypes, patients with Oligo (IDH-mut&1p/19q-codeleted) have the longest, and patients with GBM (IDH-wt) the shortest, median overall survival (OS) [7]. 

There are data suggestive of glioma subtype-specific vasculature patterns [3,8]. Glioblastoma is an end-stage tumor in which the overexpression of hypoxia- and angiogenesis-related genes have resulted in sarcomatous vascular structures as well as thrombosis and recanalization [9]. Cha et al. reported differences in the vasculature patterns of blood vessels between oligodendroglioma and astrocytoma [10]. Oligodendroglioma is known to have a unique “chicken-wire” vasculature pattern, while astrocytoma microvasculature more often displays similar features to the normal brain parenchyma.

In recent years, imaging markers derived from clinical or advanced magnetic resonance imaging (MRI) have increasingly been researched to provide insight into tumor vascularization of the whole tumor in a non-invasive manner. This is of particular interest diagnostically in tumors not displaying contrast enhancement because conventional features of angiogenesis are lacking. Vessel size imaging (VSI) is an emerging MRI technique that creates a quantitative map of vessel size, which estimates the mean diameter of vessels in each voxel of the brain MRI scan, in addition to a cerebral blood volume (CBV) map [11,12]. The VSI technique utilizes the dynamic acquisition of both T2-weighted and T2*-weighted images, to follow a bolus of gadolinium-based contrast agent (GBCA) as it passes through the vasculature. Studies have shown that T2-weighted sensitivity peaks for capillary-sized vessels and is used for measuring the microvascular cerebral blood volume (μCBV), while T2*-weighted sensitivity plateaus over a broad range of vessel sizes and has been widely used for measuring the well-known perfusion parameter of CBV [13]. Chakhoyan et al. showed that vessel size and CBV measured with the VSI technique had a good agreement with histopathological measurements of vessel caliber and vessel density in high-grade glioma [14].

While many studies have investigated the value of CBV in predicting tumor angiogenesis, tumor grade, molecular profiles, and overall survival in glioma [15,16,17,18], very few studies have explored the potential of vessel size quantification in revealing the vascular characteristics of different glioma subtypes [8,19]. Further, so far and to the best of our knowledge, no study has explored the VSI technique in differentiating microvasculature features in the three most common adult-type diffuse glioma subtypes classified based on the 2021 WHO classification of tumors of the central nervous system: Astro (IDH-mut), Oligo (IDH-mut&1p/19q-codeleted), and GBM (IDH-wt) [6]. This study aimed to explore the glioma lineage-specificity of the blood vessels by VSI, with a particular focus on non-enhancing tumors when angiogenesis is not yet visible on conventional imaging. Additionally, we performed a histopathology analysis of tumor vasculature with the endothelial cell marker CD31 to compare the MRI-based microvascular features in the three glioma subtypes with their histopathological counterparts. 

## 2. Materials and Methods

### 2.1. Patients

A prospectively collected dataset from the iGENE study, consisting of 44 patients aged 18 years or older, was used for the current analysis. The iGENE study aimed to assess the value of advanced MRI for diagnosing non-enhancing molecular subtypes of adult-type diffuse glioma. This study was approved by the Institutional Review Board of Erasmus MC, with all patients providing informed written consent to have their information stored and used for research prospectively and retrospectively (MEC-2016-717). All MRI scans were acquired preoperatively, typically one or two days before the surgery. Only patients diagnosed with glioma, confirmed by histopathology after tumor resection or biopsy, were included in the current work. Visual inspection of pre- and postcontrast T1-weighted imaging was performed by an experienced neuroradiologist to ensure that the tumor was indeed non-enhancing. Histopathological examination and molecular subclassification of 1p/19q codeletion status and IDH mutation status were performed as part of the diagnostic routine. The study was initiated prior to the publication of the WHO 2021 classification, but all molecular data were available to classify each tumor based on the WHO 2021 criteria. We excluded 6 patients due to incomplete data or insufficient image quality; therefore, the MRI analysis was performed in 38 patients. Additionally, 21 patients were included for histopathological analysis by an expert neuropathologist who determined that the hematoxylin and eosin (H&E)-stained slides of prepared tissue samples contained sufficient tumor cells. 

### 2.2. MRI Acquisition

All patients underwent 3T MRI scanning (GE, Milwaukee, WI, USA) prior to surgery with a 32-channel head coil. To obtain the microvessel parameters, Hybrid Echo Planar Imaging (HEPI) was used to simultaneously collect both T2-weighted images with spin echo (HEPI-SE) and T2*-weighted images with gradient echo (HEPI-GRE). The acquisition parameters of the HEPI scan were: TR of 1500 ms; TE(GRE)/TE(SE) of 18.6/69 ms; voxel size of 1.9 × 1.9 × 4.0 mm^3^; slice thickness of 3.0 mm (with 1.0 mm gap between slices); 120 time points; partial brain coverage with 15 slices placed over the tumor [20]. HEPI was performed with the administration of 7.5 mmol of GBCA (Gadovist, Bayer, Leverkusen, Germany), 20 TRs after the start of the HEPI acquisition. A pre-load bolus of equal size was given 5 min prior to the HEPI scan. The MRI protocol also included high-resolution structural images as part of routine clinical imaging. These were T1-weighted (TE/TR: 2.1/6.1 ms; TI: 450 ms; voxel size: 1.0 × 1.0 × 0.5 mm^3^), T2-weighted (TE/TR: 107/10,000 ms; voxel size: 0.5 × 0.5 × 3.2 mm^3^), and T2-weighted FLAIR (TE/TR: 106/6000 ms; TI: 1890 ms; voxel size: 0.6 × 0.5 × 0.5 mm^3^), all with the field of view covering the entire brain. T1-weighted scans were collected both before GBCA injection (precontrast T1-weighted) and after injection of the pre-load bolus but prior to the HEPI scan (postcontrast T1-weighted). A diffusion-weighted scan, used for estimation of the apparent diffusion coefficient (ADC) required for vessel size measurements, was also included in the protocol with acquisition parameters of TE/TR of 63/5000 ms; voxel size of 1.0 × 1.0 × 3.0 mm^3^; 3 b-values of 0, 10, 1000 s/mm^2^; and with the field of view covering the entire brain. 

### 2.3. MR Image Pre-Processing

The first four brain volumes of both HEPI-GRE and HEPI-SE were removed to ensure that the signal had reached a steady state. If visually detected, motion artifacts were corrected for both HEPI-GRE and HEPI-SE data, using MC FLIRT (FMRIB’s Linear Image Registration Tool, University of Oxford, Oxford, UK), rigidly registering all brain volumes to the fifth brain volume of HEPI-SE [21,22]. 

In-house code developed in Python 3.6 (http://www.python.org) (accessed on 4 April 2022) was used for acquiring vessel size and CBV maps. Signal–time curves were plotted for each voxel for both HEPI-GRE and HEPI-SE data separately. We excluded voxels exhibiting a drop of fewer than 2 standard deviations of the baseline signal due to the contrast agent passing through, in both HEPI-GRE and HEPI-SE data, to minimize voxels with erroneous results [23]. Then, HEPI-GRE and HEPI-SE signal–time curves were converted to transverse relaxation rates of $∆R_{2}^{*}\left(t\right)$ and ∆R_2_(t), respectively, using the following equations [13]: (1)$$∆R_{2}^{*}\left(t\right)=-\left(1⁄{\mathrm{TE}}_{\mathrm{GRE}}\right)\times \left(\ln\left(S_{\mathrm{GRE}}\left(t\right)/S_{\mathrm{GRE}}\left(0\right)\right)\right),$$
(2)$$∆R_{2}^{\;}\left(t\right)=-\left(1⁄{\mathrm{TE}}_{\mathrm{SE}}\right)\times \left(\ln\left(S_{\mathrm{SE}}\left(t\right)/S_{\mathrm{SE}}\left(0\right)\right)\right),$$
where TE is the echo time; S_GRE_(t) and S_SE_(t) are the signal–time curves of HEPI-GRE and HEPI-SE. Similarly, S_GRE_(0) and S_SE_(0) are the baseline signal of the S_GRE_(t) and S_SE_(t) for the time prior to the GBCA bolus arrival. 

The CBV and µCBV maps were estimated voxelwise from $∆R_{2}^{*}\left(t\right)$ and ∆R_2_(t), respectively, by measuring the trapezoidal integration in the time interval of entrance time (t0) to exit time (t1) of the GBCA bolus: (3)$$\mathrm{CBV}=\int_{t_{0}}^{t_{1}}∆R_{2}^{*}\left(t\right){\mathrm{dt}}^{\;},$$
(4)$$\text{µ}\mathrm{CBV}=\int_{t_{0}}^{t_{1}}∆R_{2}^{\;}\left(t\right){\mathrm{dt}}^{\;},$$

Normalization of CBV and µCBV maps was performed by dividing all voxels by the mean CBV and µCBV values of normal-appearing white matter voxels in the hemisphere contralateral to the tumor (NAWM-contra) [12]. We will refer to the normalized CBV and µCBV as the CBV and µCBV throughout the rest of the paper.

The estimation of mean vessel size for each voxel was based on the Kiselev model and was obtained by the following equation [12]:(5)$${\mathrm{Vessel}\;\mathrm{Size}}_{\;}=0.867{\left(\mathrm{CBV}\;\times \;\mathrm{ADC}\right)}^{1/2}\left[\frac{∆R_{2}^{*}}{{\left(∆R_{2}^{\;}\right)}^{3/2}}\right],$$

CBV used in Equation (5) was scaled to the median value of CBV in NAWM-contra of 3.2% [24]. ADC is the apparent diffusion coefficient generated from the diffusion-weighted scan and obtained directly from GE scanner (software version of DV25.0_R01_1451.a). Both $∆R_{2}^{*}$ and ∆R_2_ represent the maximum change in the transverse relaxation rates of $∆R_{2}^{*}\left(t\right)$ and ∆R_2_(t), respectively.

It is worth mentioning that no leakage correction algorithms were used in calculating the CBV, µCBV, or vessel size in this study. This is because no or very little contrast agent leakage was observed in the postcontrast T1-weighted image in the tumor area, as per the inclusion criteria; thus, GBCA leakage is not a cause of concern in the current analysis [17]. The in-house developed algorithm of Glioseg was applied on the pre postcontrast T1-weighted, postcontrast T1-weighted, T2-weighted, and T2-weighted FLAIR images to generate a 3D tumor mask (Figure 1A). This automatic tumor segmentation was performed by five different algorithms [25,26,27,28] (details described in Appendix A). 

A normal-appearing white matter mask from the hemisphere contralateral to the tumor (NAWM-contra) was extracted for CBV and µCBV normalization, as described above. To obtain the NAWM-contra mask, first, FAST in FSL (FMRIB’s Automated Segmentation Tool, Version 6.0.5) was applied on the brain-extracted precontrast T1-weighted image to generate probability maps of white matter, grey matter, and cerebrospinal fluid [29]. The white matter probability map from the hemisphere contralateral to the tumor was then manually selected, and, to minimize partial volume effects, binarized with a probability threshold of 0.9 and eroded with FSLMATHS in FSL using a kernel size of 3 × 3 × 3 mm^3^. 

Tumor and NAWM-contra masks were then registered to the HEPI space. For this, the brain-extracted precontrast T1-weighted image was linearly registered to the fifth brain volume of HEPI-SE (after preprocessing) using Elastix (version 4.8) [30]. Then, the same transformation matrix was used to transfer the NAWM-contra and tumor masks to the HEPI-SE space (Figure 1B).

### 2.4. Histopathological Analysis

Histology slides produced from paraffin-embedded tumor biopsies were routinely stained for H&E and adjacent cuts were immunohistochemically stained for CD31, a marker for endothelial cells. For the virtual microscopy analysis, the slides were scanned with the Hamamatsu Nanozoomer 2.0-HT C9600-12 (Hamamatsu Photonics, Hamamatsu City, Shizuoka Prefecture, Japan) to obtain whole-slide high-resolution images. 

The H&E-stained slides of the prepared tissue samples were examined by an experienced neuropathologist for tumor content. For the histopathological analysis, two regions of interest (ROIs), representative of vasculature features in the whole tissue, were selected by the experienced neuropathologist for each tissue sample. In these ROIs, all positively stained endothelial cells and vessel structures were segmented and quantified based on the CAIMAN algorithm (http://www.caiman.org.uk) (accessed on 2 February 2022) [31]. The quantification included vessel density, average vessel radius, and average vessel roundness for each patient. Vessel density was estimated by measuring the area of the stained vessels in both ROIs divided by the total area of the ROIs. Each vessel radius in the ROIs was measured with the CAIMAN algorithm and averaged across the ROIs. Vessel roundness is a metric describing how elongated the object is together with how smooth or crenated the external boundary of each vessel is. A roundness of 1 reflects a vessel with a completely circular shape, while a higher value of this parameter is associated with a more irregular vessel shape. 

### 2.5. Statistical Analysis

Mean and median values of MRI-derived CBV, µCBV, and vessel size were computed in the tumor area for each patient. A hot spot approach was also used where, initially, clusters of voxels in the 90th percentile within the tumor mask were identified. The values within these clusters were averaged and reported for each parameter. Mean, median, and hot spot values were measured across all clusters together and were then averaged for each of the three glioma subtypes separately. Additionally, the histology-derived parameters of vessel density, vessel radius, and vessel roundness were first measured for each tissue sample in the two predefined regions of interest (ROIs). Then, these parameters were group-averaged for each glioma subtype. It is noteworthy that selected ROIs were quite small and homogenously stained; thus, only the mean of the histology-derived parameters was obtained in each ROI.

All statistical analyses were performed with the statistics package of SciPy (version 1.10.1) in python version 3.6. For group comparison, first, the normality of both the MRI-derived and histology-derived parameters of each subtype was examined with the Shapiro–Wilk test. For normally distributed data, the one-way analysis of variance test (ANOVA) was used. If the group means were significantly different among glioma subtypes, a post hoc Tukey honestly significant difference (HSD) test was performed to identify the groups which are responsible for the significant variation. For non-normally distributed data, the Kruskal–Wallis (KW) test was used and, if it yielded a statistically significant result, it was followed by a post hoc Dunn’s test using a Bonferroni correction. A *p*-value of 0.05 or lower was considered significant in all tests.

## 3. Results

### 3.1. Patient Characteristics

Thirty-eight patients (median age of 42 years; age range 22–77 years; 25 men) were included, in 21 of whom (median age of 42 years; age range 28–71 years; 17 men) sufficient tissue was available for the histopathological analysis. Eighteen patients were diagnosed with Oligo (IDH-mut&1p/19q-codeleted), 13 with Astro (IDH-mut), and 7 with Glioblastoma (IDH-wt) (Table 1). 

### 3.2. MRI-Based Measurements

The group averages of CBV, µCBV, and vessel size for each of the three glioma subtypes are provided in Table 2. Mean and median measurements of CBV, µCBV, and vessel size were significantly different between the three subtypes (*p* < 0.05). Hot spot measurements showed a significant difference between the three subtypes only for µCBV. 

Pairwise comparison of mean CBV and mean µCBV showed a significant difference between Oligo (IDH-mut&1p/19q-codeleted) and Astro (IDH-mut) (*p* = 0.004 and *p* < 0.001, respectively), with Oligo (IDH-mut&1p/19q-codeleted) having significantly higher values for both these parameters. 

In addition, µCBV showed a trend of being higher in GBM (IDH-wt) compared to Astro (IDH-mut); however, this did not reach the statistically significant level (*p* = 0.12) (Figure 2).

A pairwise comparison of mean vessel size revealed a significantly lower value for GBM (IDH-wt) in comparison to Astro (IDH-mut) (*p* = 0.01). Although there was a trend for Oligo (IDH-mut&1p/19q-codeleted) to have a smaller mean vessel size compared to Astro (IDH-mut) (*p* = 0.08), this difference did not reach the statistical significance level (Figure 2c). No difference was observed in CBV, µCBV, and vessel size between Oligo (IDH-mut&1p/19q-codeleted) and GBM (IDH-wt) (*p* = 0.43, *p* = 0.73, and *p* = 0.42, respectively). 

Figure 3 illustrates exemplary slices of CBV, µCBV, and vessel size maps for a representative patient of each subtype. In Oligo (IDH-mut&1p/19q-codeleted) and GBM (IDH-wt), both CBV and µCBV are distinctly high in the tumor area, while vessel size maps show low-to-intermediate intensity in the tumor area. In Astro (IDH-mut), the reverse is seen: CBV and µCBV are evidently low in the tumor area, while the vessel size map displays several hot spots in the tumor area.

### 3.3. Histology-Derived Measurements

The results of the histopathological analyses are shown in Figure 4. There was a significant difference in vessel density between the three subtypes (*p* = 0.03); however, the pairwise comparison only showed a trend of higher vessel density in GBM (IDH-wt) compared to Oligo (IDH-mut&1p/19q-codeleted) (*p* = 0.06). No statistically significant difference was found in vessel radius between each of the three glioma subtypes. Vessel morphology, however, was significantly different in GBM (IDH-wt) compared to both Oligo (IDH-mut&1p/19q-codeleted) (*p* = 0.02) and Astro (IDH-mut) (*p* = 0.03), with GBM (IDH-wt) having the least round-shaped microvasculature.

## 4. Discussion 

In this study, we determined the microvascular characteristics of the three adult-type diffuse glioma subtypes Astro (IDH-mut), Oligo (IDH-mut&1p/19q-codeleted), and GBM (IDH-wt) showing no contrast-enhancement on MRI. Even in the absence of contrast-enhancement—a characteristic macroscopic feature of angiogenesis—there was a significant difference between the three glioma subtypes in MRI-based CBV, μCBV, and vessel size measurements. Moreover, histopathological analysis of the tumor vasculature with the endothelial cell marker CD31 suggested that the morphology of microvasculature in the GBM (IDH-wt) subtype was different from the other two subtypes, having more elongated and crenated morphology. Oligo (IDH-mut&1p/19q-codeleted) showed significantly higher CBV and μCBV compared to Astro (IDH-mut). This finding is in accordance with previous studies [15,16]. Both subtypes are IDH-mutant glioma, but the microvascular proliferation in 1p/19q-codeleted tumors is more common than in 1p/19q intact tumors [32]. Moreover, the Oligo (IDH-mut&1p/19q-codeleted) subtype is typically characterized by small, branching, chicken wire-like blood vessels in histology [10]. This pattern was also observed in most of the stained tissue slides of Oligo (IDH-mut&1p/19q-codeleted) (Figure 5) and may therefore underlie the MRI findings, as the susceptibility contrast measured with CBV is not only sensitive to vessel density, but also to the size, shape, and orientation of the blood vessels [8,10,11].

The quantitative tissue analysis performed with the CAIMAN algorithm, however, showed hardly any difference in microvascularization between the Oligo (IDH-mut&1p/19q-codeleted) and Astro (IDH-mut) glioma subtypes. A possible explanation for this seemingly contradictory finding is that the CAIMAN algorithm segments and counts stained objects in the slices that are not necessarily blood vessels. Previous studies have reported that while the CD31 antibody is useful for marking tumor angiogenesis, it can also stain non-vascular cells within soft tissue tumors and lead to errors in the analysis [33]. The GBM (IDH-wt) subtype showed a significantly smaller vessel size, as measured with MRI compared to Astro (IDH-mut), a finding that may be of use in clinical practice to non-invasively identify this most aggressive glioma subtype before surgery. In the literature, the reported microvasculature features in GBMs have been conflicting so far. While in many studies dilated glomeruloid-type vessels and microvascular proliferation are linked with GBM [34], it has been shown that vessels in these tumors may be similar to those in the normal brain parenchyma [35]. However, the vessel lumina of high-grade glioma may also narrow because of increasing numbers of their cellular constituents, due to microvascular proliferation [36]. Our histopathological findings support the latter in demonstrating that in GBM (IDH-wt) was the greatest loss of circular shapes of the vessels compared to Astro (IDH-mut) and Oligo (IDH-mut&1p/19q-codeleted), which is characteristic of GBM and potentially the cause of finding a decreased vessel size on MRI. It is worth noting that here, we focused on non-enhancing GBM (IDH-wt), which is clinically highly relevant as only a small proportion of GBMs (IDH-wt) do not enhance and may mimic lower grade tumors. Our findings, which may seem contradictory to some of the previous literature, may be thus related to the selection of these tumors. Additionally, we used the molecular classification for diagnosing GBM (IDH-wt) rather than histopathology, which removes the circularity between the empirical (tissue) measurements and the ground truth, which for histopathological diagnosis includes vessel characteristics. Again, this could also underline differences between our findings and those reported in the (older) literature.

We do know from the literature that IDH wild-type glioma correlates with an overexpression of angiogenesis-related genes, eventually resulting in increased perfusion [9]. Our study showing a trend for both CBV and μCBV, as measured with MRI, as well as vessel density measured in tissue to be higher in GBM (IDH-wt) compared to Astro (IDH-mut), is in line with this finding of overexpressed angiogenesis-related genes. That these differences did not reach the statistically significant level might be explained by the small sample size (*n* = 7) in combination with the large observed heterogeneity of these parameters in GBM (IDH-wt) [15]. Again, it should be noted here that the GBM (IDH-wt) subtype included in the present study was non-enhancing lesions and therefore less typical for the GBMs that come with extensive microvascular proliferation. 

Our study had some limitations. The iGENE study was designed to investigate non-enhancing gliomas, which are typically lower grade gliomas; therefore, a smaller number of patients were diagnosed with GBM (IDH-wt). The current work is therefore of an exploratory nature because of the relatively small, unequally distributed sample size. However, our findings can serve as a good starting point for the validation of VSI in a larger cohort to investigate whether VSI could provide an effective tool for the diagnosis of non-enhancing adult-type diffuse glioma subtypes. Another limitation of this study is that not all patients’ tissue samples were available for histopathological analysis. Finally, the biopsy location was unknown for the included samples; thus, regional heterogeneity within the tumors could not be accounted for in the histopathological analysis. However, the tissue findings were broadly in line with the MRI-based measurements.

## 5. Conclusions

In summary, this exploratory work evaluated how VSI can provide information about the microvasculature features of the three subtypes of non-enhancing adult-type diffuse glioma. From a clinical perspective, our findings are of potential interest for differentiating the most aggressive tumor subtype, GBM (IDH-wt), from the Astro (IDH-mut) subtype, requiring prompt treatment amongst this subset of non-enhancing tumors with otherwise low-grade appearance. Our findings offer direction for future work aimed at confirming these findings in a larger validation cohort, preferably with targeted biopsies, to more precisely correlate MRI findings with tissue characteristics. 

## Figures and Tables

**Figure 1 cancers-15-02135-f001:**
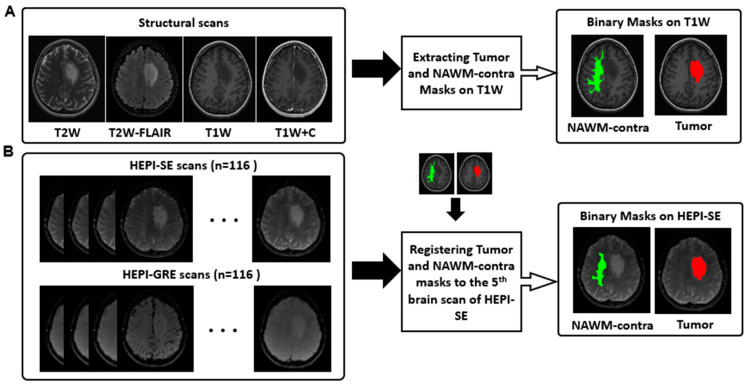
Preprocessing workflow. (**A**) T2-weighted (T2W), T2-weighted FLAIR (T2W-FLAIR), precontrast T1-weighted (T1W) and postcontrast T1-weighted (T1W + C) images are used for extracting binary masks of tumor (highlighted in red) and NAWM-contra (highlighted in green); (**B**) brain scans of HEPI-GRE and HEPI-SE; the generated binary masks of tumor (highlighted in red) and NAWM-contra (highlighted in green) are transferred from structural space to HEPI space. The overlaid NAWM-contra and tumor masks have a numerical value of one, with distinct colors employed solely to enhance visual differentiation.

**Figure 2 cancers-15-02135-f002:**
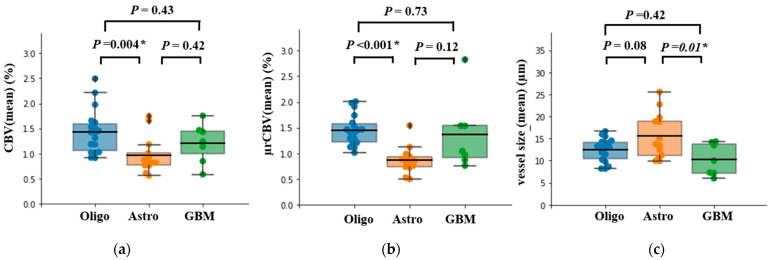
Box plot of MRI-derived measurements of (**a**) CBV, (**b**) µCBV, and (**c**) vessel size, with mean bar for the three glioma subtypes Oligo (IDH-mut &1p/19q codeleted), Astro (IDH-mut), and GBM (IDH-wt). Tukey HSD test or Dunn’s test using a Bonferroni correction was used for pairwise statistical analysis. ‘*’ indicates significant difference between two subtypes accompanied by the exact *p*-value.

**Figure 3 cancers-15-02135-f003:**
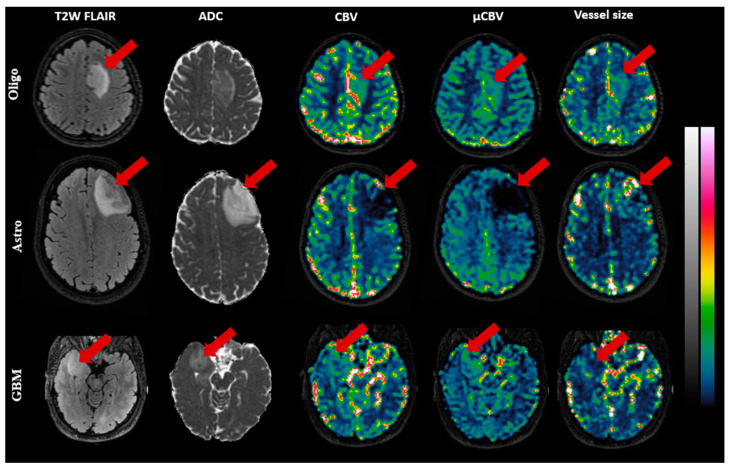
Exemplary MR images of three patients, Oligo (IDH-mut &1p/19q co-deleted), Astro (IDH-mut), and GBM (IDH-wt), respectively, from top to bottom. The images (with color bars on the right) from left to right, respectively, are: T2W-FLAIR, ADC (0–1 µm^2^/s), CBV(0–10%), µCBV(0–10%), and vessel size (0–100 µm), with the red arrow pointing to the tumor.

**Figure 4 cancers-15-02135-f004:**
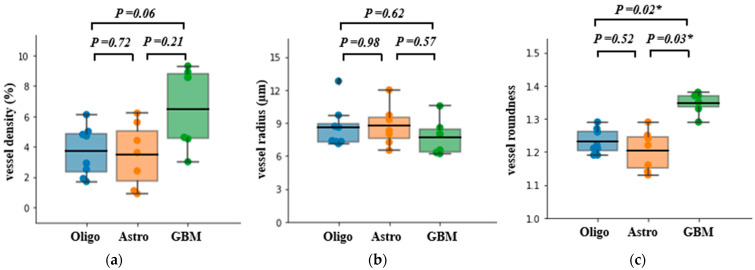
Box plot of histology-derived measurements: (**a**) vessel density, (**b**) vessel radius, and (**c**) vessel roundness, with mean bar for the three glioma subtypes Oligo (IDH-mut &1p/19q codeleted), Astro (IDH-mut), and GBM (IDH-wt). Tukey honestly significant difference (HSD) test or Dunn’s test using a Bonferroni correction was used for pairwise statistical analysis. ‘*’ indicates a significant difference between two subtypes accompanied by the exact *p*-value.

**Figure 5 cancers-15-02135-f005:**
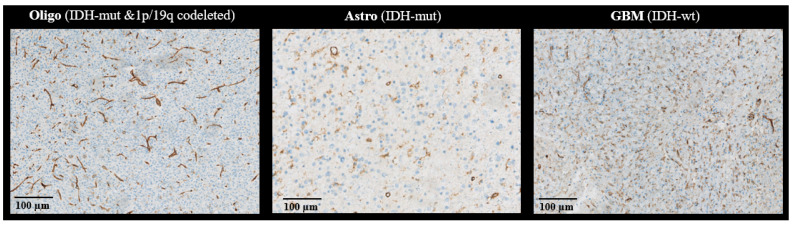
Exemplary CD31-stained tissue of three patients with Oligo (IDH-mut &1p/19q co-deleted), Astro (IDH-mut), and GBM (IDH-wt), from left to right.

**Table 1 cancers-15-02135-t001:** Patient characteristics and histopathological grade based on the WHO 2021 classification of tumors of the central nervous system [6]. Patient characteristics with additional histopathological tissue analysis in brackets.

Tumor Subtype	Grade 2	Grade 3	Grade 4	Total	Sex	Age (Years)
Oligo (IDH-mut&1p/19q-codeleted)	15	3	-	18	13M/5F	40 ± 12
	(6)	(2)	-	(8)	(7M/1F)	(43 ± 13)
Astro (IDH-mut)	9	4	-	13	5M/8F	36 ± 8
	(7)	(-)	(-)	(7)	(4M/3F)	(35 ± 6)
GBM (IDH-wt)	-	-	7	7	7M/0F	60 ± 10
	(-)	(-)	(6)	(6)	(6M/0F)	(59 ± 8)

**Table 2 cancers-15-02135-t002:** Group averages of CBV, µCBV, and vessel size in the three glioma subtypes: Oligo (IDH-mut&1p/19q-codeleted), Astro (IDH-mut), and GBM (IDH-wt). ^a^: one-way ANOVA test; ^b^: Kruskal–Wallis test. The bold formatting indicates a significant *p*-value.

		Oligo (IDH-mut&1p/19q-codeleted)	Astro (IDH-mut)	GBM (IDH-wt)	*p*-Value
**MRI-derived parameters**					
CBV (%)	Mean (std)	1.43 (0.44)	0.96 (0.36)	1.20 (0.39)	**0.01 ^a^**
	Median (std)	1.17 (0.36)	0.70 (0.18)	0.92 (0.37)	**<0.001 ^b^**
	Hot spot (std)	3.31 (1.14)	2.79 (1.71)	3.34 (0.85)	0.13
µCBV (%)	Mean (std)	1.45 (0.29)	0.86 (0.26)	1.36 (0.71)	**<0.001 ^b^**
	Median (std)	1.31 (0.25)	0.74 (0.21)	1.29 (0.73)	**<0.001 ^b^**
	Hot spot (std)	2.37 (0.60)	1.62 (0.77)	2.22 (0.83)	**0.02 ^a^**
Vessel size (μm)	Mean (std)	12.48 (2.61)	15.55 (5.10)	10.33 (3.62)	**0.01 ^a^**
	Median (std)	10.18 (2.02)	10.91 (2.47)	7.02 (2.32)	**0.002 ^a^**
	Hot spot (std)	44.91 (15.55)	78.85 (46.48)	53.81 (34.96)	0.06 ^a^
**Histopathology-derived parameters**					
Vessel density	Mean (std)	3.69 (1.64)	3.45 (2.09)	6.47 (2.74)	**0.03 ^a^**
Vessel radius (μm)	Mean (std)	8.61 (1.92)	8.72 (1.82)	7.68 (1.70)	0.55 ^b^
Vessel roundness	Mean (std)	1.23 (0.03)	1.20 (0.06)	1.34 (0.03)	**<0.001 ^a^**

## Data Availability

Data are available on request.

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
