# Peer review of "Microvasculature Features Derived from Hybrid EPI MRI in Non-Enhancing Adult-Type Diffuse Glioma Subtypes"

_cancers, 2023, doi:10.3390/cancers15072135_

Round 1

Reviewer 1 Report

The specificity of VSI to differentiate glioma subtype is promising. This would provide valuable insights into therapeutic of GBM. 

This paper is presented in a logic and informative way. Their conclusions were supported by their data.

I only have one minor question. Please indicate the meaning of color bar in fig1.

Author Response

Dear Reviewer1,

Thank you for your kind words and positive feedback on our work. We greatly appreciate your support and encouragement. We're delighted to hear that you found our methodology and findings to be well-presented and informative.

In response to your question regarding the color bar meaning in Figure 1, we have updated the figure caption in the resubmitted manuscript to include this information. The updated figure caption can be found in the resubmitted manuscript.

Best regards,
Fatemeh

Reviewer 2 Report

Thanks a lot for your invitation to review this manuscript "Microvasculature features derived from hybrid EPI MRI in 2 non-enhancing adult-type diffuse glioma subtypes" (cancers-2284270). disclosed some interesting results on tumor microvasculature. I agree with its publication in the current version. The subject of this manuscript is interesting and is within the scope of the journal.

Author Response

Dear Reviewer2,

Thank you for your positive feedback and kind words about our work. We appreciate your support and encouragement, and we are thrilled to hear that you found the subject of the manuscript interesting and relevant to the scope of the journal.

Best regards,

Fatemeh

Reviewer 3 Report

This exploratory work evaluated how vessel size imaging via MRI can provide information about microvasculature features of the three subtypes of non-enhancing adult-type gliomas. The histopathology-based analysis showed that high grade gliomas possessed vessels with more irregular shapes than the two other subtypes.  From a clinical perspective, the findings are of potential interest for differentiating the most aggressive tumor subtype from less aggressive lower grade tumors based on MRI findings.  It is suggestive that this information may be helpful in directing tumor management.  It is anticipated that these findings will correlate with the pathology of targeted biopsies.  The findings in this study are of some possible application in the management of brain tumors, although the clinical implications remain speculative. 

Author Response

Dear Reviewer3, 

I would like to express my sincere appreciation for your time and effort in reviewing our manuscript. We are honored that you found our work useful in the management of brain tumors. 

We agree that more research is needed to fully understand the clinical implications of our findings. However, we believe understanding the microvasculature in different subtypes of gliomas can provide valuable information for designing and implementing treatments that target the blood vessels that feed the tumor, potentially improving treatment outcomes and overall survival for patients.

We have considered your comments in regard to correcting the minor spelling issues that you highlighted. Please find the updated version of our manuscript attached. 

Best regards,

Fatemeh
